# Grip power test: A new valid and reliable method for assessing muscle power in healthy adolescents

**Álvaro Huerta Ojeda**[1]*, **Blanca Fontecilla Díaz**[2], **María Mercedes Yeomans Cabrera**[3], **Daniel Jerez-Mayorga**[4]

**1** Grupo de Investigación en Salud, Actividad Física y Deporte ISAFYD, Universidad de Las Américas, Santiago, Chile, **2** Magíster Medicina y Ciencias del Deporte, Facultad de Ciencias, Universidad Mayor, Santiago, Chile, **3** Facultad de Educación, Universidad de Las Américas, Santiago, Chile, **4** Faculty of Rehabilitation Sciences, Universidad Andres Bello, Santiago, Chile

\* achuertao@yahoo.es

**Funding:** The authors received no specific funding for this work.

**Competing interests:** The authors have declared that no competing interests exist.

## Abstract

The assessment of the strength and muscle mass of the hand-finger segment are reliable indicators of health and predictors of cardiometabolic risk in the adult population. However, there are no valid and reliable tests to assess the muscle power of this segment in healthy adolescents. The objective of this study was to determine the validity and inter-day reliability of a grip power test (Grip$_W$ test) in healthy adolescents. Twenty-one adolescents (15.61 ± 2.20 years old) were part of the study. All participants were instructed to perform a grip with incremental load sets from 1–10 kg as fast as possible. The validity of the Grip$_W$ test was determined with the load-power curve and linear regression equation. Inter-day reliability considered the coefficient of variation (CV), intra-class correlation coefficient (ICC), and standard error of the mean (SEM). The significance level for all statistical analyses was $p < 0.05$. The parabola in the load-power curve for both hands showed normality for the Grip$_W$ test. In addition, the analysis showed a CV = 4.63% and ICC = 1.00 for the right hand, while the left hand showed a CV = 3.23% and ICC = 1.00. The Grip$_W$ test proved to be valid and reliable for assessing gripping muscle power functionally and unilaterally in healthy adolescents.

## Introduction

The ability of humans to manipulate objects is unique [1]. Our fingers, along with the ability to adapt to various textures and surfaces, can exert different force levels [2]. Likewise, the strength of the hand-finger segment, called grip strength, is recognized as a predictor of risk for several pathologies [3], including nutritional status imbalance, level and quality of muscle mass [4], and even psychosocial disturbances such as anxiety [5]. Indeed, grip strength is a reliable indicator of the population's physical function [4] and health status [6]. Therefore, to reduce mortality risk, grip strength requires accurate diagnosis and appropriate intervention [7].

The most commonly used test to evaluate grip strength is the handgrip test [4]. Thus, various measuring instruments have been used to assess the strength of the hand-finger segment,

ranging from hydraulic dynamometers [8] to isokinetic dynamometers [9]. This indicator has been evaluated in all age groups, regardless of the protocol or instruments used to assess grip strength [6, 8, 10]. In this regard, Bohannon et al. [6] consider that grip strength is a relevant biomarker of health in the elderly. At the same time, Alahmari et al. [8] used grip strength as an indicator of health in the adult population. Likewise, Huerta-Ojeda et al. [5] related low levels of grip strength with anxiety-trait in university women. However, despite this evidence, Fredriksen et al. [10] concluded that grip strength should not be used as a tool to detect cardiometabolic risk factors in prepubertal children.

As shown, grip strength is a reliable indicator of the population's physical function and health status [4, 6]. Indeed, upper extremity muscle power has been assessed through bench press [11], rock climbing [12], or seated ball throwing, both for older [13] and trained individuals [14]. However, the tests that evaluate the muscular power of the upper extremities consider global body movements [11, 12, 14] and not specific segments such as the hand-finger segment. This situation conditions that the power of the hand-finger segment is determined through the non-specific relation of one of its components (power of the upper extremities) [15]. As previously stated, only grip strength [4] and muscle mass [16] would correspond to this body segment since most of the existing upper extremity power tests are global and do not necessarily represent the hand-finger segment [11, 13, 14]. Indeed, the validation of a test to measure the power of the hand-finger segment will complement functional tests to determine muscle quality in healthy adolescents. For this reason, and to make it possible to determine the muscle power of the hand-finger segment, a specific test for grip power (Grip$_W$ test) should be validated.

According to the above, the main objective of this study was to determine the validity and inter-day reliability of a Grip$_W$ test in adolescents aged 11–19 years.

## Materials and methods

### Approach to the study

All study participants attended the laboratory five days at 48-hour intervals. During the first visit, basic anthropometric assessments were performed. The second and third days were devoted to familiarizing the participants with the power test for the gripping musculature and the assessment of manual grasp through the handgrip test. On the fourth and fifth days, the power of the grip musculature was assessed through the handgrip test. The inclusion criterion was that all study participants were free of skeletal muscle injuries in the upper extremities during testing. In addition, to avoid a drop in the participants' performance, they were instructed not to engage in physical activity during the weeks of evaluations. All the tests proposed in the study were performed unilaterally, starting with the dominant extremity of the participants.

### Design

A repeated-measures design was used to compare the test-retest inter-day reliability for different power and velocity variables collected during Grip$_W$ test exercises.

### Participants

The sample consisted of 21 physically active adolescents of both sexes who voluntarily agreed to be part of the study (convenience sample). All participants read and signed an informed assent, while their guardians performed the same procedure with an informed consent. The study, assent and informed consent were approved by the Ethics Committee of the

Universidad Mayor (registration number 193–2020) and under the latest version of the Helsinki declaration [17].

## Anthropometry

Body mass was determined using a digital scale (TERRAILLON®, Hong Kong, China), height was determined using a portable stadiometer (SECA 213®, Hamburg, Germany), and the BMI was determined by dividing the kilograms of weight by the square of the height in meters $(kg/m^2)$.

## Handgrip test

Before starting the test, each participant performed a standardized warm-up of dynamic movements of the upper extremities for 5 minutes. The participant was positioned standing with shoulder adducted with neutral rotation, elbow in 180˚ extension, forearm, and wrist in a neutral position [18]. For correct positioning of the angles, a goniometer (JENOR®, Valencia, Spain) was used. Then, the manual dynamometer (CAMRY EH01®, China) was placed in the participant's hand, after which the investigator indicated the word "squeeze" to begin the test and "relax" to end it. The duration of the maximum voluntary contraction for the handgrip-test execution was 3 seconds (s). Participants performed the test twice, consecutively, with each hand (first the right hand and then the left hand). There was a 30-second pause between each repetition and a 1-minute rest before evaluating the other limb. All participants received verbal support during the execution of the test. The average result of the two repetitions per hand was used for the characterization of the sample.

## Grip power test (Grip$_W$ test)

Before starting the test, a 5-minute standardized warm-up was performed, including active mobilization of the wrist and finger joints. Each participant was positioned standing, facing forward, and laterally to the supports (in this support, the dumbbell and the linear encoder were placed). The participants grasped the dumbbell with the hand to be evaluated. Specifically, the participants set the center of the dumbbell between the interphalangeal joints of the middle and ring fingers and the distal interphalangeal joints of the index and little fingers. For the execution of the test, the participants' shoulders had to be relaxed, the elbow of the limb to be evaluated had to be extended to 180˚, and the wrist had to be placed in a neutral position (Fig 1).

Afterward, once the participants were positioned correctly, they were asked to perform finger flexion movements with thumb opposition, generating the handgrip. This movement displaced the dumbbell vertically. As part of the protocol and to create the individual load-power curve, participants were asked to perform the upward action as fast as possible. Also, participants were asked not to include shoulder, elbow, and wrist movements in the execution of the test. If this occurred, the repetition or series was considered invalid. A member of the research team carried out the qualitative evaluation of each performance. Also, all participants received verbal support during the execution of the test. It was established to start the action with the word "up" and end with the word "rest" to facilitate the development of the test and standardize muscle contraction times.

The protocol considered series with incremental loads. The loads weighed 1, 2, 3, 4, 5, 6, 7, 8, 9, and 10 kg, always in that order. The pause between series was 30 seconds, while the rest to evaluate the other extremity was one minute. The number of repetitions per set was adjusted according to the participant's capacity. The protocol considered: four repetitions per set when the participant moved the loads between 60–50 cm·s$^{-1}$; three repetitions between 49–30 cm·s$^{-1}$;

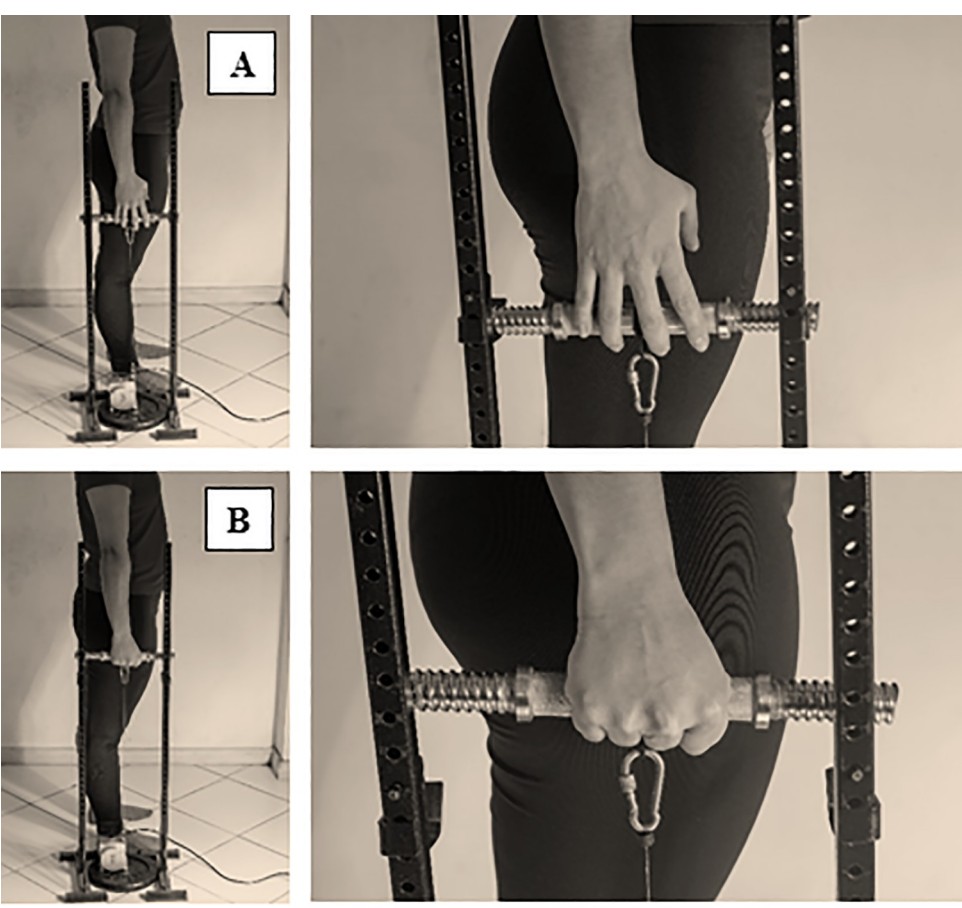

A: initial position; B: final position

**Fig 1. Grip$_w$ test of the grip muscle.** A: Initial position; B: Final position.

two repetitions between 29–20 cm·s$^{-1}$; and one repetition between 19–1 cm·s$^{-1}$ [19]. Running speed and power were evaluated through a linear encoder (CHRONOJUMP®) and software (CHRONOJUMP®, Version 1.4.6.0, Barcelona, Spain). The mean velocities and mean powers of all repetitions per load were recorded. These were averaged if the participant performed more than one repetition in the series (2–4 repetitions).

The maximum individual power was obtained by constructing a load-power curve and the respective linear regression equation (y = ax + b). The average powers of each load (1–10 kg) were used to construct this curve. The maximum powers resulting from the load-power curve and the respective linear regression equation were calculated for the test and retest, allowing the determination of the inter-day reliability of the power test for the gripping musculature.

## Data analysis

Descriptive data are presented as mean and standard deviation. The normal distribution of the data was confirmed by the Shapiro-Wilk test (p > 0.05). The concurrent validity of the Grip$_W$ test was determined with the load-power curve and the respective linear regression equation for all participants. The Student's t-test was used to determine the differences in power generated between the two sexes. Test-retest inter-day reliability was assessed through the coefficient

of variation (CV), intra-class correlation coefficient (ICC), standard error of the mean (SEM), and the corresponding 95% confidence interval. Acceptable inter-day reliability was determined as a CV < 10% and ICC > 0.85 [20]. The criteria for interpreting the strength of the *r* coefficients were as follows: trivial (<0.1), small (0.1–0.3), moderate (0.3–0.5), high (0.5–0.7), very high (0.7–0.9), or practically perfect (>0.9) [21]. The t-tests for related samples and the Bland-Altman technique were used to evaluate the concordance between the test and the retest of the $Grip_W$ test [21]. Inter-day reliability assessments were performed using a customized spreadsheet [22]. All other statistical analyses were performed with SPSS version 22.0 software (SPSS, Chicago, IL, USA). The significance level for all statistical analyses was p < 0.05.

## Results

The participants had the following characteristics: 15 men, age = 15.86 ± 2.47 years (range: 14–17), height = 1.70 ± 0.08 m (range: 1. 55–1.84), body mass = 65.99 ± 6.37 kg (range: 54.4–79.0), Body Mass Index (BMI) = 22.94 ± 2.55 $kg/m^2$ (range: 19.5–27.7), right-hand grip strength = 36.11 ± 9.20 kg (range: 22.2–51.9), and left-hand grip strength = 34.22 ± 10.35 kg (range: 17.7–49.8). 6 women, age = 15.00 ± 1.26 years (range: 14–17), height = 1.63 ± 0.06 m (range: 1. 53–1.73), body mass = 59.40 ± 9.45 kg (range: 51.2–76.5), Body Mass Index (BMI) = 22.45 ± 3.22 $kg/m^2$ (range: 19.0–28.3), right-hand grip strength = 26.60 ± 6.94 kg (range: 18.7–36.8), and left-hand grip strength = 24.20 ± 6.65 kg (range: 16.0–34.0).

The powers generated in all the loads, both in the test and the retest, did not show significant differences between men and women (p > 0.05). The power report is shown in Fig 2. After these results, it was decided to perform reliability calculations with the 21 cases as a whole.

Only the 1-kg load for both segments (right and left hand) and the 10 kg load for the left hand showed an ICC lower than 0.85 between test and retest. Likewise, only the extremes of the load-power curves for both segments showed a CV > 10% (Table 1 and Fig 3).

Concerning inter-day reliability, a CV of 4.63% (mean test = 11.12 ± 8.53, mean retest = 11.25 ± 8.53, p = 0.43; 95% IC = 0.13, SEM = 0.52, ICC = 1.00) and 3.23% (mean test = 10.32 ± 8.37, mean retest = 10.30 ± 8.36, p = 0.87; 95% IC = -0.02, SEM = 0.33, ICC = 1.00) for right- and left-hand gripping musculature were observed, respectively. At the same time, the linear regression for both segments reported a $r^2$ = 0.99 (p < 0.0001). Inter-day reliability results are reported in Fig 4.

## Discussion

This study was designed to determine the validity and inter-day reliability of a $Grip_W$ test in adolescents aged 11–19 years. The main findings of this study suggest that the parabola described in the load-power curve, both in the individual curves and in the general curve of the right and left hand, showed normality for this physical capacity [19]. Likewise, the results obtained for CV and ICC showed a high consistency between the test-retest and test-retest [21]. These findings suggest that the $Grip_W$ test is valid and reliable for assessing grip power in adolescents aged 11–19 years.

### $Grip_W$ test functionality

In addition to the above, the $Grip_W$ test allows the determination of gripping muscle power unilaterally and functionally since the participants were evaluated standing and with a hand-grip grip, which simulates daily living activities [1]. Also, there are other validated tests to assess upper extremity power unilaterally [23, 24]. Such is the case of the test proposed by Negrete et al. [23]; these researchers evaluated upper extremity power in adults aged 18–45

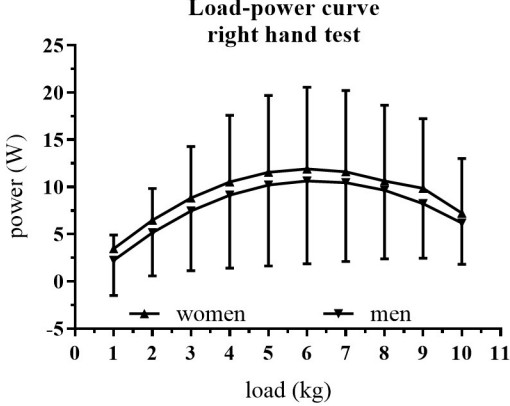

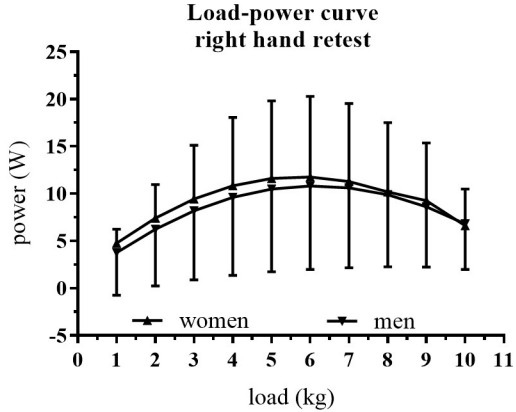

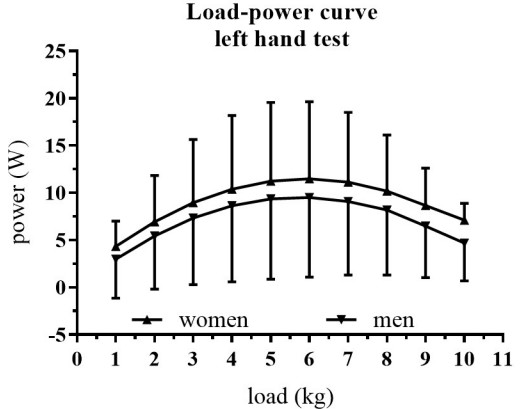

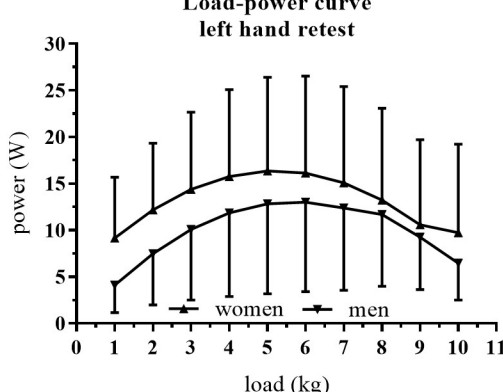

**Fig 2. Load-power curves of the grip muscles between men and women.**

years. The protocol considered throwing a 2.7 kg medicine ball from the seated position with both the dominant and non-dominant arm in this research. At the end of the study, the researchers reported an ICC of 0.98 and 0.97 for the dominant and non-dominant arm test-retest, respectively. Similarly, Riemann et al. [24] validated a 1, 2, and 3 kg medicine ball throwing test in adults aged 20–30 years; in this study, the protocol considered throws from the seated and overhead position. Although the evidence shows the existence of tests to assess upper extremity power in different populations [23, 24], these tests were poorly functional. Among other conditions, the type of grip and the protocol of both Negrete et al. [23] and Riemann et al. [24] do not simulate activities performed in daily life. In addition, the protocols used in these tests are, to a greater extent, for sports practice [25]. Based on the described background and to the best of our knowledge, the $Grip_W$ test is the first test developed to assess, in a functional manner, the power of the gripping musculature in adolescents aged 11–19 years.

## Bilateral analysis and $Grip_W$ test

In parallel, evidence shows the existence of other validated tests to evaluate upper extremity power [11, 23]; however, the protocols of these tests assess this physical capacity bilaterally. In this sense, Clemons et al. [11] evaluated upper extremity power through bench press in adults aged 20–27 years. In this research, the participants performed bench press repetitions at the maximum possible speed. These bench press loads were 25 kg for women and 61.4 kg for men.

**Table 1. Values per load and segment of the Grip_W test.**

| | | | | | | |
|---|---|---|---|---|---|---|
| **Right Hand (n = 21)** | | | | | | |
| **Exercise** | **test (W)** | **retest (W)** | **p** | **Δ** | **CV** | **ICC** |
| | **mean ± DS** | **mean ± DS** | | **95% IC** | **95% IC** | **95% IC** |
| **Load 1: 1 kg** | 2.60 ± **3.26** | 4.05 ± 3.88 | 0.04 | 1.46 | 66.19 | 0.64 |
| | | | | 0.04–2.87 | 50.64–95.58 | 0.30–0.84 |
| **Load 2: 2 kg** | 5.55 ± 4.24 | 6.58 ± 5.36 | 0.03 | 1.04 | 24.31 | 0.92 |
| | | | | 0.09–1.98 | 18.60–35.11 | 0.80–0.96 |
| **Load 3: 3 kg** | 7.86 ± 5.98 | 8.55 ± 6.77 | 0.03 | 0.69 | 12.11 | 0.98 |
| | | | | 0.05–1.33 | 9.26–17.48 | 0.95–0.99 |
| **Load 4: 4 kg** | 9.55 ± 7.41 | 9.96 ± 7.82 | 0.08 | 0.41 | 7.64 | 0.99 |
| | | | | -0.07–0.89 | 5.85–11.04 | 0.98–1.00 |
| **Load 5: 5 kg** | 10.59 ± 8.28 | 10.80 ± 8.42 | 0.29 | 0.21 | 5.87 | 0.99 |
| | | | | -0.20–0.61 | 4.49–8.48 | 0.99–1.00 |
| **Load 6: 6 kg** | 11.01 ± 8.55 | 11.09 ± 8.54 | 0.62 | 0.08 | 4.64 | 1.00 |
| | | | | -0.25–0.41 | 3.56–6.72 | 0.99–1.00 |
| **Load 7: 7 kg** | 10.79 ± 8.22 | 10.81 ± 8.19 | 0.85 | 0.02 | 3.77 | 1.00 |
| | | | | -0.24–0.28 | 2.89–5.45 | 0.99–1.00 |
| **Load 8: 8 kg** | 9.94 ± 7.32 | 9.97 ± 7.36 | 0.83 | 0.04 | 5.91 | 0.99 |
| | | | | -0.34–0.42 | 4.52–8.52 | 0.99–1.00 |
| **Load 8: 9 kg** | 8.64 ± 6.05 | 8.78 ± 6.16 | 0.70 | 0.14 | 13.39 | 0.97 |
| | | | | -0.63–0.91 | 10.18–19.56 | 0.92–0.99 |
| **Load 10: 10 kg** | 6.45 ± 4.63 | 6.77 ± 5.51 | 0.61 | 0.32 | 30.00 | 0.83 |
| | | | | -0.99–1.63 | 22.81–43.81 | 0.62–0.93 |
| **Left Hand (n = 21)** | | | | | | |
| **Exercise** | **test (W)** | **retest (W)** | **p** | **Δ** | **CV** | **ICC** |
| | **mean ± DS** | **mean ± DS** | | **95% IC** | **95% IC** | **95% IC** |
| **Load 1: 1 kg** | 3.38 ± 3.77 | 3.98 ± 4.41 | 0.06 | -0.03 | 26.92 | 0.95 |
| | | | | 0.61–1.24 | 20.60–38.88 | 0.88–0.98 |
| **Load 2: 2 kg** | 5.88 ± 5.37 | 6.31 ± 5.59 | 0.07 | 0.43 | 12.03 | 0.98 |
| | | | | -0.04–0.90 | 9.20–17.37 | 0.96–0.99 |
| **Load 3: 3 kg** | 7.80 ± 6.83 | 8.07 ± 6.79 | 0.16 | 0.28 | 7.81 | 0.99 |
| | | | | -0.12–0.68 | 5.98–11.28 | 0.98–1.00 |
| **Load 4: 4 kg** | 9.14 ± 7.83 | 9.28 ± 7.66 | 0.41 | 0.14 | 5.87 | 0.98 |
| | | | | -0.21–0.49 | 4.49–8.48 | 0.96–0.99 |
| **Load 5: 5 kg** | 9.90 ± 8.29 | 9.92 ± 8.13 | 0.86 | 0.02 | 4.35 | 1.00 |
| | | | | -0.25–0.30 | 3.33–6.28 | 0.99–1.00 |
| **Load 6: 6 kg** | 10.08 ± 8.19 | 10.00 ± 8.15 | 0.52 | -0.08 | 3.80 | 1.00 |
| | | | | -0.32–0.17 | 2.90–5.48 | 1.00–1.00 |
| **Load 7: 7 kg** | 9.68 ± 7.54 | 9.52 ± 7.75 | 0.43 | -0.16 | 6.65 | 0.99 |
| | | | | -0.57–0.25 | 5.09–9.61 | 0.98–1.00 |
| **Load 8: 8 kg** | 8.79 ± 6.51 | 8.61 ± 7.15 | 0.64 | -0.18 | 13.70 | 0.97 |
| | | | | -0.97–0.61 | 10.42–20.01 | 0.93–0.99 |
| **Load 8: 9 kg** | 7.18 ± 5.03 | 7.36 ± 6.11 | 0.73 | 0.18 | 21.69 | 0.93 |
| | | | | -0.90–1.25 | 16.39–32.08 | 0.83–0.97 |
| **Load 10: 10 kg** | 5.45 ± 3.59 | 5.96 ± 5.22 | 0.58 | 0.50 | 44.75 | 0.70 |
| | | | | -1.42–2.43 | 33.06–69.26 | 0.33–0.89 |

SD: standard deviation; W: watts; Δ: delta of variation; CV: coefficient of variation; ICC: intra-class correlation coefficient; CI: 95% confidence intervals; *p*: p-value.

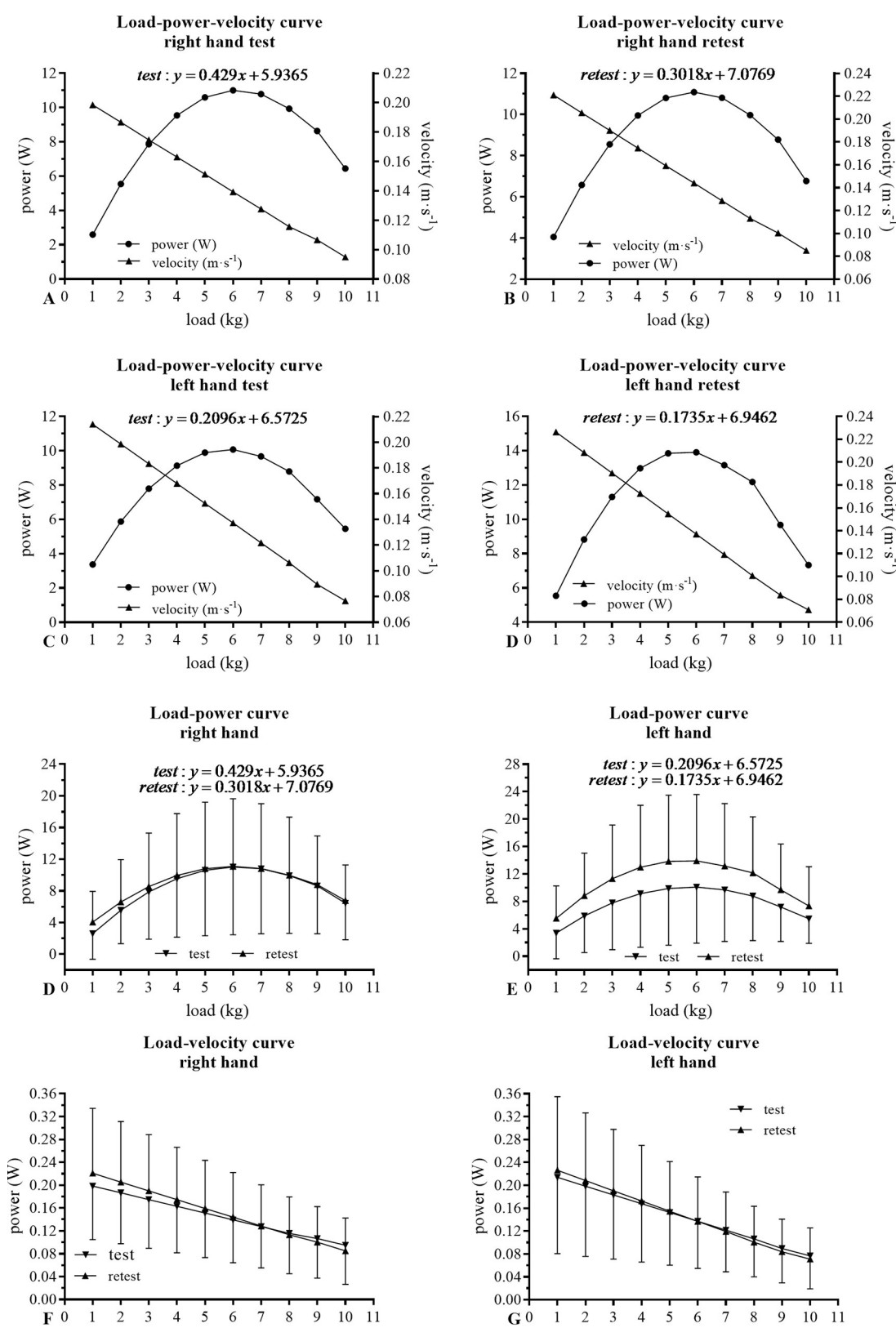

**Fig 3. Load-power and load-velocity curves of the grip muscles in test and retest.**

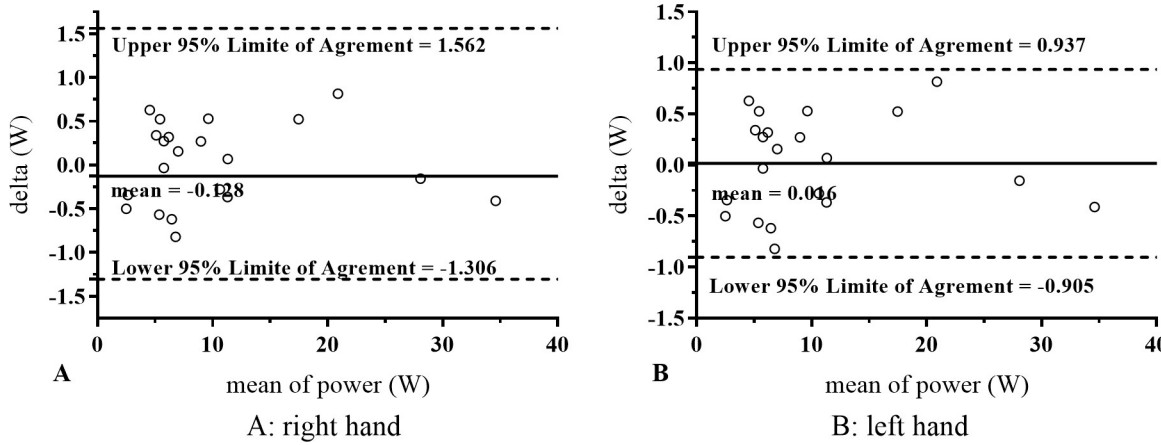

**Fig 4. The solid line represents the mean difference between the test and retest of the Grip$_W$ test (watts).** The dashed lines represent the upper and lower 95% confidence limits.

After the bench press, women threw a 6 kg medicine ball and men a 9 kg medicine ball (both medicine ball throws should be performed in a seated position on a 45° inclined bench). Likewise, Harris et al. [13] validated a test to estimate power in adults over their 70s. The protocol considered medicine ball throws of 1.5 and 3.0 kg in a seated position. In turn, Negrete et al. [23] validated a test for adults aged 18–45 years. This protocol includes performing as many push-ups and modified pull-ups repetitions as possible (supine position with a smith press) for 15 seconds. Despite the validity of previous tests, the protocols performed bilaterally do not differentiate bilateral muscle decompensation (asymmetry) in the different age groups evaluated [26]. Therefore, the Grip$_W$ test is the first test that makes it possible to assess the asymmetry of the gripping musculature.

## Specificity of the hand-finger segment of the Grip$_W$ test

Likewise, the Grip$_W$ test explicitly evaluates the muscle power of the hand-finger joint segment of both the right and left extremities. In this sense, most of the tests to assess the power of the hand-finger segment involve other joint segments, such as the shoulder, elbow, and wrist, and even the lower extremities; this last segment is used as an accessory movement to boost the mobility of the upper extremities [12, 27]. In research developed by Dhahbi et al. [27], upper extremity power was evaluated through a climbing test in adults of the special military command; this protocol considers climbing a 5-meter rope as fast as possible. In turn, Dhahbi et al. [27] evaluated grip strength with dynamometry, obtaining significant results between the 5-meter rope climbing test and the grip strength of the right and left hand ($p < 0.05$). Similarly, Laffaye et al. [12] evaluated muscle to power through explosive pull-ups in beginner and elite athletes. Although they consider muscle power, these protocols do not allow isolation of the hand-finger joint segment, preventing crossover or relationship with the handgrip test [4] or anthropometric parameters [16] of the same body segment. Likewise, the low functionality of the protocols described above contains their application in the non-athlete population. Therefore, the Grip$_W$ test is the first test that allows power assessment of the hand-finger joint segment.

## Grip$_W$ test as a component of muscle quality

The Grip$_W$ test, together with strength (assessed by the handgrip test) and muscle mass (assessed by anthropometry), allows us to obtain a specific and precise assessment of the

muscle quality of the upper extremities, specifically of the hand-finger segment [15]. However, although anthropometric evaluations of the upper extremities were conducted in the present study, an analysis to determine muscle quality was not performed since the study had a different purpose. Therefore, assessing muscle quality by calculating only the handgrip and anthropometric parameters—such as forearm length [28], hand length [8], or body composition [29] —could be unspecific and imprecise. Consequently, to obtain a specific and accurate assessment of muscle quality, we suggest using strength, power, and anthropometry tests for each body segment.

## Inter-day reliability of the Grip$_W$ test

The results also showed that the Grip$_W$ test has a high degree of inter-day reliability, both for the right segment and the left segment. These inter-day reliability data evidence a higher consistency for assessing the power of the gripping musculature than other tests used to assess the muscle power of the same body segment [12, 27]. In this sense, Laffaye et al. [12] validated a new power test in climbers; this test consisted of assessing the speed of movement with isoinertial accelerometers placed in the middle of the athletes' lower back. At the end of the study, the researchers reported a CV < 5% and an ICC = 0.98 between test-retest [12]. Similarly, Dhahbi et al. [27] validated a 5-m rope-climbing test, declaring it as a commando-specific power test of the upper limbs; at the end of the study, the investigators reported a CV < 10% and an ICC = 0.99 between test-retest. Despite these good inter-day reliability indicators presented by Laffaye et al. [12] and Dhahbi et al. [27], these studies were developed for novice and elite athletes and military commandos, respectively. Thus, the Grip$_W$ test is the first reliable test that assesses the grip power in healthy adolescents.

## Limitations

At first, it was not easy for participants to understand the Grip$_W$ test's execution. However, once the required movement familiarization sessions were completed, the participants performed well during the trial. In addition, all those repetitions that included other joint segments, such as shoulder, elbow, and wrist, were excluded. Similarly, there was difficulty in performing the movement with the non-dominant limb. However, there were no significant differences between the two segments.

At the same time, the number of participants in the study could be considered a limitation. However, we observed that the number of participants in published articles with similar methodological designs was less than 20 cases to validate different tests [30, 31]. Therefore, we considered that 21 cases were an appropriate number to validate and determine the inter-day reliability of the Grip$_W$ test.

Also, all participants in this study were physically active. However, the level of physical activity, as a product of COVID-19, was not as desired [32]. This situation may have affected the absolute value of muscle power but in no way affects the validity and inter-day reliability of the Grip$_W$ test.

## Conclusions

In conclusion, Grip$_W$ is a valid test to assess the power of the gripping musculature. Likewise, the Grip$_W$ test presents high inter-day reliability to evaluate the muscular power of the hand-finger segment of both the right and left extremities. Finally, according to the protocol developed, the Grip$_W$ test is the first test designed to evaluate, functionally and unilaterally, the power of the gripping muscles in adolescents aged 11–19 years.

## Practical applications

For researchers and trainers using the Grip$_W$ test, it is recommended to consider the following aspects: 1, before applying the test, participants should have a familiarization period; in this phase, researchers and trainers should correct the location of the hand and fingers. Then, participants should be informed that a correct execution includes a handgrip movement at the highest possible speed. 2, educate about recurring errors during execution, such as the inclusion of other joint segments like shoulder, elbow, and wrist. 3, consider the level of the physical condition of the participants since a poor physical condition could affect the inter-day reliability of the test.

We suggest that the Grip$_W$ test be applied as another indicator of health since, together with grip strength and determination of forearm muscle mass, it would allow estimating the muscular quality of the hand-finger segment. Likewise, the Grip$_W$ test can be used in sports with the dominant use of the gripping musculature such as judo, wrestling, or climbing.

## Acknowledgments

We thank the 21 participants who voluntarily attended the various days on which the Grip$_W$ test procedures were performed.

## Author Contributions

**Conceptualization:** Álvaro Huerta Ojeda, Blanca Fontecilla Díaz, María Mercedes Yeomans Cabrera, Daniel Jerez-Mayorga.

**Data curation:** Álvaro Huerta Ojeda.

**Formal analysis:** Álvaro Huerta Ojeda, Blanca Fontecilla Díaz.

**Investigation:** Álvaro Huerta Ojeda, Blanca Fontecilla Díaz.

**Methodology:** Álvaro Huerta Ojeda, Blanca Fontecilla Díaz, Daniel Jerez-Mayorga.

**Project administration:** Álvaro Huerta Ojeda, Blanca Fontecilla Díaz.

**Resources:** Álvaro Huerta Ojeda.

**Software:** Álvaro Huerta Ojeda.

**Supervision:** Álvaro Huerta Ojeda.

**Validation:** Álvaro Huerta Ojeda, Blanca Fontecilla Díaz.

**Visualization:** Álvaro Huerta Ojeda, María Mercedes Yeomans Cabrera, Daniel Jerez-Mayorga.

**Writing – original draft:** Álvaro Huerta Ojeda, Blanca Fontecilla Díaz, María Mercedes Yeomans Cabrera, Daniel Jerez-Mayorga.

**Writing – review & editing:** Álvaro Huerta Ojeda, Blanca Fontecilla Díaz, María Mercedes Yeomans Cabrera, Daniel Jerez-Mayorga.

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
