## [Decision Letter · Decision Letter 0]

26 Jul 2021

PONE-D-21-18765

Grip Power Test: a New Valid and Reliable Method for Assessing muscle power in healthy adolescents

PLOS ONE

Dear Dr. Huerta Ojeda,

Thank you for submitting your manuscript to PLOS ONE. After careful consideration, we feel that it has merit but does not fully meet PLOS ONE’s publication criteria as it currently stands. Therefore, we invite you to submit a revised version of the manuscript that addresses the points raised during the review process.

ACADEMIC EDITOR:

Dear Authors,

two reviewers expert in the filed revised your ms and found some major points you should reply.

Please take into consideration only the methodological suggestions the reviewers pointed out.

We look forward to receiving your revised manuscript.

Kind regards,

Emiliano Cè

Academic Editor

PLOS ONE

Journal Requirements:

3. We note that you have stated that you will provide repository information for your data at acceptance. Should your manuscript be accepted for publication, we will hold it until you provide the relevant accession numbers or DOIs necessary to access your data. If you wish to make changes to your Data Availability statement, please describe these changes in your cover letter and we will update your Data Availability statement to reflect the information you provide

Reviewers' comments:

Reviewer's Responses to Questions

**Comments to the Author**

1. Does the manuscript provide a valid rationale for the proposed study, with clearly identified and justified research questions?

Reviewer #1: Yes

Reviewer #2: Yes

2. Is the protocol technically sound and planned in a manner that will lead to a meaningful outcome and allow testing the stated hypotheses?

Reviewer #1: Partly

Reviewer #2: Partly

3. Is the methodology feasible and described in sufficient detail to allow the work to be replicable?

Reviewer #1: No

Reviewer #2: No

4. Have the authors described where all data underlying the findings will be made available when the study is complete?

Reviewer #1: Yes

Reviewer #2: Yes

5. Is the manuscript presented in an intelligible fashion and written in standard English?

Reviewer #1: Yes

Reviewer #2: Yes

6. Review Comments to the Author

You may also provide optional suggestions and comments to authors that they might find helpful in planning their study.

Reviewer #1: The present study aimed to determine the validity and reliability of a Grip W test in healthy adolescent since this test has been evaluated in other age groups such as elderly, adult population, university women and prepubertal children.

To test their hypothesis the authors used a sample of 21 adolescents that in my opinion is too small to validate and determine the reliability of the Grip W test even if in literature there were 2 studies that did the same. In addition, in order to test the validity and reliability of the Grip W test, I think is important to consider the gender differences and employ a sample balanced in the sex distribution. In your study you used a sample of 15 men and 6 women and you reported all the results together without distinctions. As a limitation, the authors declared that the participants reported being less physically active due to the COVID-19 pandemic scenario and I think that the fitness condition of the subject is an important aspect to consider when you tried to validate a test. So, I was wondering if your participants were active or inactive since I think is important to specify it when you validate a test. Furthermore, I don’t understand some points of the methodology. For example, how much time the participants must squeeze the manual dynamometer during the handgrip test? How much time the participants must displace the dumbbell vertically during the grip w test? In addition, adjust the number of repetitions per set according to the participant’s capacity and after that present the results all together without distinctions is not the right option to validate a methodology.

Reviewer #2: Huerta Ojeda et al.- Grip Power Test: a New Valid and Reliable Method for Assessing muscle power in healthy adolescents

Overview: The present study assessed the validity and reliability of grip power test in adolescents. I think that this work illustrates an argument very interesting, especially in a practical point of view: it concerns grip test, argument already and extensively studied, but it adds a novel test in a specific population as element of newness. However, there are some points that need some clarification and revision. See my specific comments below.

Lines 65-66: A brief paragraph about the importance and the purposes of GripW test) should be added. Please, specify in all paper what is the reliability assessed (inter-day, intra-day)

Line 88: Please, add the sample size calculation

Line 90: I suggest checking SD of the stature

Lines 92-93: I suggest replacing this information in result paragraph

Lines 105-106: Please, specify the warm-up procedures? Were they standardized or individualized?

Lines 106-108: Was the position choose based on a reference?

Lines 111-112: Was the handgrip test performed with both hands? (Please, specify before lines 112) If so, were randomized? Was performed a familiarized test? During the test was the participant verbally encouraged? Should the operator influence the measure?

Lines 111-112: Have the procedures been taken by some references?

Lines 116-117: Please, specify the warm-up procedures? Were they standardized or individualized?

Lines 130-131: How the authors verified this “Also, participants were asked not to include shoulder, elbow, and wrist movements in the execution of the test”?

Lines 150-151: Please, specify the reliability assessment if was intra-day or inter-day. It was not clear also considering the experimental protocol above written. Were the tests conducted at the same time?

Line 154: “are presented as means and standard deviations” should be “are presented as mean and standard deviation”

Line 155: Please, specify the criteria to validity assess.

Line 160: Please, report all the classification for CV and ICC

Line 162: Please, add a reference for r coefficients

Table 1: Please, specify what the authors meant with Load 1, 2, 3,,,,. Moreover, what the authors reported in tables 1 as test and retest? How they explained these higher SDs almost the same of mean?

Lines 186-188: The sentence in lines 184-186 should test the validity but not the reliability. Please, add a sentence for reliability results.

Line 248: Please, specify hot has been determined the muscle mass? To assess the quality of the muscle it is needed other measurements. Moreover, have been determined such as forearm length, hand length? Please, review the entire paragraph.

Lines 257-258: I suggest avoiding data in the discussion part.

Lines 259-260: Please, review accuracy and reliability terms because in these lines were used as synonymous

7. PLOS authors have the option to publish the peer review history of their article (what does this mean?). If published, this will include your full peer review and any attached files.

Reviewer #1: **Yes: **Letizia Galasso

Reviewer #2: No

---

## [Author Response · Author response to Decision Letter 0]

16 Aug 2021

Dear Editor, 

One of the suggestions made by reviewer #1 was the inclusion of comparison between genders. To comply with this request, the research group decided to include a figure in the manuscript (Figure 2).

This situation increases the total number of Tables and Figures to 5 (one table and four figures). If the article is accepted for publication, we hope that you and the Journal would allow the inclusion of Figure 2.

Dear Reviewer 1

We are writing to provide an answer to the suggestions given. The modifications are marked in bold type in the manuscript. Also, in this letter, we present the modifications performed and/or answers to each question.

To test their hypothesis the authors used a sample of 21 adolescents that in my opinion is too small to validate and determine the reliability of the GripW test even if in literature there were 2 studies that did the same. In addition, in order to test the validity and reliability of the GripW test, I think is important to consider the gender differences and employ a sample balanced in the sex distribution. In your study you used a sample of 15 men and 6 women and you reported all the results together without distinctions. As a limitation, the authors declared that the participants reported being less physically active due to the COVID-19 pandemic scenario and I think that the fitness condition of the subject is an important aspect to consider when you tried to validate a test. So, I was wondering if your participants were active or inactive since I think is important to specify it when you validate a test. Furthermore, I don’t understand some points of the methodology. For example, how much time the participants must squeeze the manual dynamometer during the handgrip test? How much time the participants must displace the dumbbell vertically during the grip w test? In addition, adjust the number of repetitions per set according to the participant’s capacity and after that present the results all together without distinctions is not the right option to validate a methodology.

Answer: 

As a research group, we know that 21 adolescents are a somewhat low sample. However, as described in the manuscript, we relied on articles published in other journals. Fortunately for us, the load-velocity-power curves for this specific muscle group had similar kinetics to other muscle groups. This antecedent, for the research group, and hopefully for the reviewer's view, allows us to validate the GripW test to measure grip muscle power.

In addition, as requested by the reviewer, an analysis by gender was added. Furthermore, this analysis (information related to the characterization of the sample) was relocated to the "results" section. Regarding the analysis by gender, before performing the reliability calculations of the GripW test, the research group calculated the differences between men and women. On that occasion, we observed no test-retest differences in both hands between men and women. This antecedent led us to decide to perform the reliability calculations for all participants, grouped by gender. Despite this, we added a figure representing this comparison (Figure 2).

Regarding the level of physical activity of the participants, they were physically active. However, due to the COVID-19 pandemic, the level of physical activity was not the desired one (the correction was made in the manuscript).

Regarding the hand-held dynamometer grip time, this information was added to the methodology.

Concerning the time that the participants must move the dumbbell vertically in the grip test, the protocol states that “participants were asked to perform the upward action as fast as possible” (L128-129).

Concerning the number of repetitions per set in the GripW test, to define the number of repetitions, we based ourselves on "a scale of perception of velocity in resistance exercise," described by Bautista et al. (2014). This reference is described in the methodology. Although it does not make explicit the number of repetitions per movement velocity, in practice, we have realized that between 2-4 repetitions allow us to normalize and stabilize the data. For the reasons above, we decided to consider "four repetitions per set when the participant moved the loads between 60-50 cm·s-1; three repetitions between 49-30 cm·s-1; two repetitions between 29-20 cm·s-1; and one repetition between 19-1 cm·s-1”. Likewise, we believe that replicating this protocol will generate standardized results, and for this reason, we make them explicit in the methodology.

Dear Reviewer 2

We are writing to provide an answer to the suggestions given. The modifications are marked in bold type in the manuscript. Also, in this letter, we present the modifications performed and/or answers to each question.

Comment 1: Lines 65-66: A brief paragraph about the importance and the purposes of GripW test) should be added. Please, specify in all paper what is the reliability assessed (inter-day, intra-day)

Answer: A short paragraph on the importance of the GripW test was added. In addition, "inter-day reliability" was included.

Comment 2: Line 88: Please, add the sample size calculation

Answer: In the manuscript, we make it explicit that the sample was for convenience. For this reason, we do not have a sample size calculation. 

We relied on validity and reliability studies with fewer than 20 participants (doi 10.1080/17461391.2019.1704068; doi 10.1519/jsc.0000000000003118).

Comment 3: Line 90: I suggest checking SD of the stature

Answer: Corrected

Comment 4: Lines 92-93: I suggest replacing this information in result paragraph

Answer: This information was replaced in the results 

Comment 5: Lines 105-106: Please, specify the warm-up procedures? Were they standardized or individualized?

Answer: It was specified that the warm-up was standardized for all participants.

Comment 6: Lines 106-108: Was the position choose based on a reference?

Answer: Yes, the position was based on references. The reference was included.

Comment 7: Lines 111-112: Was the handgrip test performed with both hands? (Please, specify before lines 112) If so, were randomized? Was performed a familiarized test? Comment 1: During the test was the participant verbally encouraged? Should the operator influence the measure?

Answer: All comments were specified in the manuscript.

Comment 8: Lines 111-112: Have the procedures been taken by some references?

Answer: Yes, the reference was included in paragraph.

Comment 9: Lines 116-117: Please, specify the warm-up procedures? Were they standardized or individualized?

Answer: It was specified that the warm-up was standardized for all participants.

Comment 10: Lines 130-131: How the authors verified this “Also, participants were asked not to include shoulder, elbow, and wrist movements in the execution of the test”?

Answer: It was specified that a member of the research team carried out the qualitative evaluation of each execution.

Comment 11: Lines 150-151: Please, specify the reliability assessment if was intra-day or inter-day. It was not clear also considering the experimental protocol above written. Were the tests conducted at the same time?

Answer: The following was corrected throughout the manuscript: "inter-day reliability".

Comment 12: Line 154: “are presented as means and standard deviations” should be “are presented as mean and standard deviation”

Answer: Corrected

Comment 13: Line 155: Please, specify the criteria to validity assess.

Answer: It was made explicit that the validity determined was concurrent validity and that "the load-power curve and the respective linear regression equation for all participants" developed in the GripW test were used to determine it.

Comment 14: Line 160: Please, report all the classification for CV and ICC

Answer: We state that "Acceptable inter-day reliability was determined as a CV < 10% and ICC > 0.85". We believe it is unnecessary to state the entire classification based on this antecedent since the parameters are either met or not met. There is no in-between.

Comment 15: Line 162: Please, add a reference for r coefficients

Answer: Reference included.

Comment 16: Table 1: Please, specify what the authors meant with Load 1, 2, 3,,,,. Moreover, what the authors reported in tables 1 as test and retest? How they explained these higher SDs almost the same of mean?

Answer: 

Table 1 shows the mean test-retest powers for the right and left hands. This information was also corrected in Table 1.

Only the SDs of the first load, both in test and retest, were more significant than the mean values. Since there was a large dispersion among the 21 participants.

Comment 17: Lines 186-188: The sentence in lines 184-186 should test the validity but not the reliability. Please, add a sentence for reliability results.

Answer: Sentence included

Comment 18: Line 248: Please, specify hot has been determined the muscle mass? To assess the quality of the muscle it is needed other measurements. Moreover, have been determined such as forearm length, hand length? Please, review the entire paragraph.

Answer: We added explanations related to muscle quality assessment. In addition, we gave responses to comments made by the reviewer.

Comment 19: Lines 257-258: I suggest avoiding data in the discussion part.

Answer: Corrected

Comment 20: Lines 259-260: Please, review accuracy and reliability terms because in these lines were used as synonymous

Answer: The term used was corrected.

---

## [Decision Letter · Decision Letter 1]

5 Oct 2021

Grip Power Test: a New Valid and Reliable Method for Assessing muscle power in healthy adolescents

PONE-D-21-18765R1

Dear Dr. Huerta Ojeda,

We’re pleased to inform you that your manuscript has been judged scientifically suitable for publication and will be formally accepted for publication once it meets all outstanding technical requirements.

Kind regards,

Emiliano Cè

Academic Editor

PLOS ONE

Additional Editor Comments (optional):

Reviewers' comments:

Reviewer's Responses to Questions

**Comments to the Author**

1. Does the manuscript provide a valid rationale for the proposed study, with clearly identified and justified research questions?

Reviewer #1: Yes

Reviewer #2: Yes

2. Is the protocol technically sound and planned in a manner that will lead to a meaningful outcome and allow testing the stated hypotheses?

Reviewer #1: Yes

Reviewer #2: Yes

3. Is the methodology feasible and described in sufficient detail to allow the work to be replicable?

Reviewer #1: Yes

Reviewer #2: Yes

4. Have the authors described where all data underlying the findings will be made available when the study is complete?

Reviewer #1: Yes

Reviewer #2: Yes

5. Is the manuscript presented in an intelligible fashion and written in standard English?

Reviewer #1: Yes

Reviewer #2: Yes

6. Review Comments to the Author

You may also provide optional suggestions and comments to authors that they might find helpful in planning their study.

Reviewer #1: I received the revised manuscript for review. I'm glad to see that the authors took reviewers' advice into consideration. The current manuscript is better than the one before it.

Reviewer #2: I appreciate all the corrections made my the authors. The article is now well described and written.

7. PLOS authors have the option to publish the peer review history of their article (what does this mean?). If published, this will include your full peer review and any attached files.

Reviewer #1: **Yes: **Letizia Galasso

Reviewer #2: **Yes: **Marta Borrelli

---

## [Editor Report · Acceptance letter]

12 Oct 2021

PONE-D-21-18765R1 

Grip Power Test: a New Valid and Reliable Method for Assessing muscle power in healthy adolescents 

Dear Dr. Huerta Ojeda:

I'm pleased to inform you that your manuscript has been deemed suitable for publication in PLOS ONE. Congratulations! Your manuscript is now with our production department. 

Kind regards, 

on behalf of

Professor Emiliano Cè 

Academic Editor

PLOS ONE